# MeanSparse: Post-Training Robustness Enhancement Through Mean-Centered Feature Sparsification

## Abstract

We present a simple yet effective method to improve the robustness of both Convolutional and attention-based Neural Networks against adversarial examples by post-processing an adversarially trained model. Our technique, MeanSparse, cascades the activation functions of a trained model with novel operators that sparsify mean-centered feature vectors. This is equivalent to reducing feature variations around the mean, and we show that such reduced variations merely affect the model's utility, yet they strongly attenuate the adversarial perturbations and decrease the attacker's success rate. Our experiments show that, when applied to the top models in the RobustBench leaderboard, MeanSparse achieves a new robustness record of 75.28% (from 73.71%), 44.78% (from 42.67%) and 62.12% (from 59.56%) on CIFAR-10, CIFAR-100 and ImageNet, respectively, in terms of AutoAttack accuracy. Code: https://anonymous.4open.science/r/MeanSparse-84B0/

## 1 Introduction

*Adversarial Training* (AT) Madry et al. (2018) has become a popular method to defend deep neural networks (DNNs) against adversarial examples Szegedy et al. (2013); Goodfellow et al. (2014). The core idea involves generating adversarial examples during the training phase and incorporating them into the training dataset. The mere existence of such adversarial examples are due to features that are non-robust to imperceptible changes Ilyas et al. (2019). AT attenuates the significance of these non-robust features on model's output, therefore improving robustness.

While AT has demonstrated effectiveness in enhancing model robustness, it faces limitations in generalizability across various methods for generating adversarial examples and suffers from low training efficiency. Various directions have been taken Xing et al. (2022); Gowal et al. (2021); Wong et al. (2019) to overcome these challenges in order to enhance the robustness provided by AT. In this work, we explore a complementary direction towards improving AT's robustness: The design of activation functions presents a promising, yet underexplored, avenue for improving AT. Smooth Adversarial Training (SAT) has shown that replacing ReLU with a smooth approximation can enhance robustness Xie et al. (2020), and activation functions with learnable parameters have also demonstrated improved robustness in adversarially trained models Dai et al. (2022); Salimi et al. (2023a). This paper aims to further investigate the potential of activation functions in enhancing AT, providing new insights and approaches to bolster model robustness.

In this work, we explore the impact of sparsifying features to enhance robustness against adversarial examples. We introduce the MeanSparse technique, which integrates sparsity into models trained using AT. Our findings indicate that non-robust features persist in the model, albeit with reduced values, even after applying AT. This insight inspired our approach to partially block these remaining non-robust features. To achieve this, we propose imposing sparsity over mean-centered features, denoted by *Mean-based Sparsification*, inspired by sparsity-promoting regularizers commonly used for feature denoising Elad & Aharon (2006); Aharon et al. (2006) and feature selection Tibshirani (1996); Mairal et al. (2010).

The MeanSparse operator selectively suppresses variations around the mean of feature representations, effectively filtering out non-robust features. For a given feature channel, we compute the mean ($\mu$) and standard deviation ($\sigma$) over the training set. Using a tunable threshold ($Th = \alpha\sigma$), we block

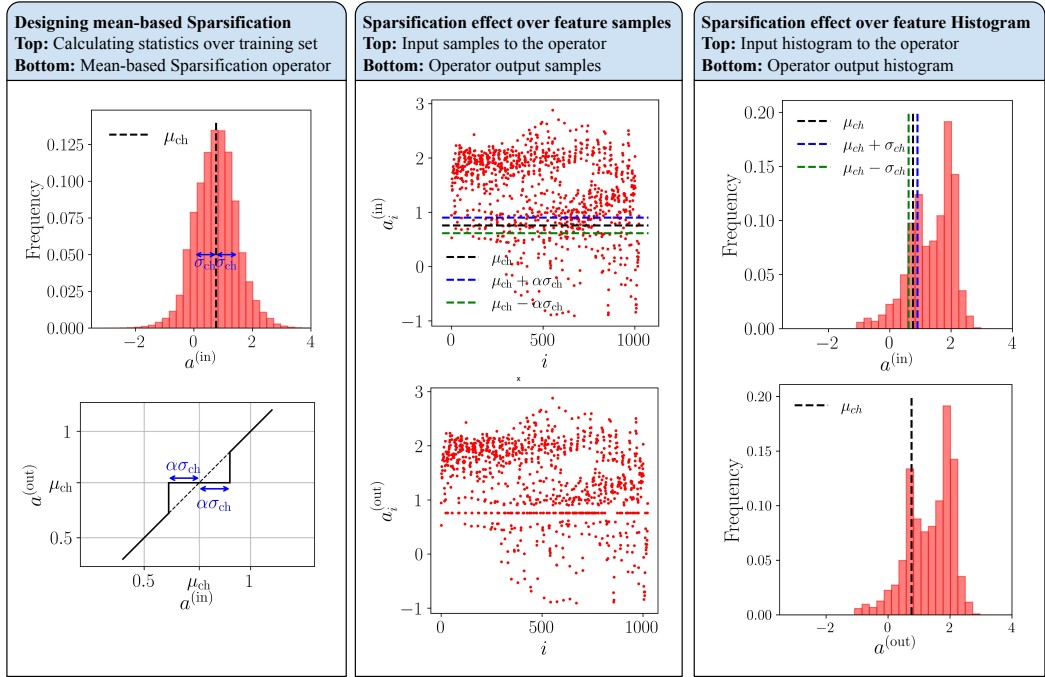

Figure 1: Mean-based sparsification operator used in the MEANSPARSE technique for hypothetical channel *ch*. The first column represents the design procedure. First, the mean ($\mu_{\text{ch}}$) and standard deviation ($\sigma_{\text{ch}}$) are calculated over the training set (top figure). The mean-based sparsification operator is designed with hyper-parameter $\alpha$ which blocks the variations in the $\alpha\sigma_{\text{ch}}$ vicinity of $\mu_{\text{ch}}$ (bottom figure). The second column represents how mean-based sparsification affects the input features for one test sample (top figure) and generates output features (bottom figure). The effect of mean-based sparsification over the feature histogram is also demonstrated in the third column.

feature values that lie within $\mu \pm Th$, replacing them with the mean value ($\mu$). This operation limits minor perturbations that adversarial attacks often exploit, while preserving the informative structure of features outside this range. For instance, consider a hypothetical feature channel with a mean ($\mu$) of 0.5 and standard deviation ($\sigma$) of 0.2. Setting $\alpha = 1$, we block values between 0.3 and 0.7, replacing them with 0.5. This simple mechanism attenuates insignificant variations, as demonstrated in Figure 1 of the paper, where we visualize how the input histogram is transformed. The blocked region corresponds to low-information variations, enhancing robustness by reducing the attacker's exploitable capacity.

Figure 1 illustrates the mean-based sparsification operator, a crucial component of the MEANSPARSE technique. In the first column, the histogram for a randomly selected channel in the first layer of the RaWideResNet trained on the CIFAR-10 dataset Peng et al. (2023) is displayed, showing the mean and standard deviation calculated over the CIFAR-10 training set (top figure). The mean-based sparsification operator (bottom figure), parameterized by $\alpha$ blocks variations around the mean. In the second column, the top figure shows the features of a hypothetical channel for a CIFAR-10 test image. The sparsification operator blocks variations between the blue and green dashed lines, producing the output shown in the bottom figure. The third column provides a similar visualization for the input and output histograms of the sparsification operator. Blocking high-probability (low information) variations prevents deterioration of model performance while limiting the attacker's ability to exploit that region.

MEANSPARSE establishes a new state-of-the-art (SOTA) in robustness. When applied to the leading robust models, it improves $\ell_\infty$ AutoAttack accuracy from 73.71% to 75.28% on CIFAR-10, from 42.67% to 44.78% on CIFAR-100, and from 59.56% to 62.12% on ImageNet, while maintaining nearly unchanged clean accuracy. Additionally, when applied to the top model for $\ell_2$ AutoAttack accuracy on CIFAR-10, MEANSPARSE boosts robust accuracy from 84.97% to 87.28%. In all

four cases, MEANSPARSE achieves the highest ranking according to RobustBench specifications Croce et al. (2020).

In summary, we make the following contributions:

- Integrating MEANSPARSE into the top robust models from the RobustBench benchmark Croce et al. (2020) improved robustness, establishing new records for $\ell_\infty$ AutoAttack accuracy: on CIFAR-10 (from 73.71% to 75.28%), CIFAR-100 (from 42.67% to 44.78%), and ImageNet (from 59.56% to 62.12%). Additionally, new $\ell_2$ AutoAttack accuracy record was set on CIFAR-10 (from 84.97% to 87.28%).
- The developed MEANSPARSE technique can be easily integrated into trained models, enhancing their robustness at almost no additional cost.
- We identified critical limitations of current activation functions and developed MEANSPARSE to mitigate these limitations. Our simulations show that regardless of the activation function, MEANSPARSE improves robustness upon integration.
- Through various experiments, we demonstrated that the improved robustness resulting from MEANSPARSE can be generalized across different model sizes and architecture types.

## 2 PRELIMINARIES

### 2.1 NOTATIONS

We use bold lowercase letters to show vectors while the $i$-th element of vector $\boldsymbol{x}$ is represented by $x_i$. Two vector norm metrics are used throughout the paper, including $\ell_0 = \|\cdot\|_0$ (number of non-zero elements) and $\ell_2 = \|\cdot\|_2$ (Euclidean norm). The $\ell_0$ norm can also effectively penalize the variables with small values but its nonsmoothness limits its application. The proximal operator finds the minimizer of function $f$ in the vicinity of input vector $\boldsymbol{v}$ and plays an essential role in our intuition to design the mean-based sparsification operator.

**Definition 2.1** (Proximal operator Parikh & Boyd (2014)). Let $f : \mathbb{R}^n \to \mathbb{R} \cup \{+\infty\}$ be a proper and lower semi-continuous function. The proximal operator (mapping) $\text{prox}_f : \mathbb{R}^n \to \mathbb{R}^n$ of $f$ at $\boldsymbol{v}$ is defined as:

$$\text{prox}_f(\boldsymbol{v}) = \arg\min_{\boldsymbol{x}} \ f(\boldsymbol{x}) + \frac{1}{2}\|\boldsymbol{x} - \boldsymbol{v}\|_2^2 \tag{1}$$

The definition of proximal operator can be extended to nonsmooth functions such as $\ell_0$ norm. Assume $f(\boldsymbol{a}) \triangleq \lambda\|\boldsymbol{a}\|_0$. Then the proximal operator can be calculated as Elad (2010):

$$\text{prox}_f(\boldsymbol{v}) = \mathcal{H}_{2\lambda}(\boldsymbol{v}), \ \mathcal{H}_\alpha(v) = \begin{cases} v & |v| > \sqrt{\alpha} \\ 0 & |v| \leq \sqrt{\alpha} \end{cases} \tag{2}$$

where $\mathcal{H}_\alpha(\cdot)$ is the element-wise hard-thresholding operator.

### 2.2 RELATED WORK

Adversarial training, a method to enhance model robustness, encounters high complexity because of the adversarial attack design incorporated during training. The concept of using adversarial samples during training was first introduced in Szegedy et al. (2013) and then AT was introduced in Madry et al. (2018). *Fast Gradient Sign Method* (FGSM) is a low-complexity attack design used for AT Tramèr et al. (2018); Liu et al. (2021); Wong et al. (2019); Vivek & Babu (2020). This approach while reducing the complexity of AT, cannot generalize to more complex adversarial attacks. On the other hand using *Projected Gradient Descent* with AT, although increasing the attack design complexity due to its iterative nature, can better generalize to other adversarial attacks Madry et al. (2018); Tramèr et al. (2018); Wang et al. (2019); Cheng et al. (2022). *Curriculum Adversarial Training* gradually increases the complexity of adversarial samples during AT Cai et al. (2018). Other types of adversarial attacks also have been proposed to be used for AT including *Jacobian-based Saliency Map Attack* Qin et al. (2019), Carlini and Wagner attack Carlini & Wagner (2017); Wen et al. (2018) and attacking Ensemble of methods during AT Pang et al. (2019), among others.

Different activation functions have been proposed that generally target to improve the generalizability of DNNs Glorot et al. (2011); Clevert et al. (2015); Ramachandran et al. (2017); He et al. (2015); Li

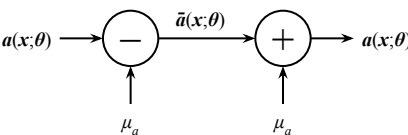

Figure 2: Mean-centered feature used in regularized optimization problem of equation 3

et al. (2024); Liang et al. (2021); Elfwing et al. (2018); Hendrycks & Gimpel (2018). Bounded ReLU (BReLU) where the output of ReLU is clipped to avoid adversarial perturbation propagation has shown robustness improvements in standard training scenarios Zantedeschi et al. (2017). The effect of symmetric activation functions Zhao & Griffin (2016), data-dependent activation functions Wang et al. (2020), learnable activation functions Tavakoli et al. (2021) and activation function quantization Rakin et al. (2018) over the robustness of the model in standard training scenario have also been explored.

The performance of different non-parametric activation functions in the AT scenario is explored in Gowal et al. (2020) and smooth activation functions demonstrated better robustness. This result is on par with SAT where the authors argued that the ReLU activation function limits the robustness performance of AT due to its non-smooth nature. In other words, the non-smooth nature of ReLU makes the adversarial sample generation harder during training. Thus they proposed the SAT which replaces ReLU with its smooth approximation and trained using Adversarial training. The results show improved robustness while preserving the accuracy Xie et al. (2020). The effect of non-parametric activation function curvature has also been studied which shows that lower curvature improves the robustness Singla et al. (2021).

Parametric activation functions increase the flexibility of activation functions while the parameters can be learned during the training phase. This flexibility, if designed intelligently, is also shown to be effective in improving robustness with AT. An example of Parametric activation functions is parametric shifted SiLU (PSSILU) Dai et al. (2022). Similar to Neural Architecture Search (NAS), *Searching for Adversarial Robust Activation Functions* (SARAF) is designed to search over candidate activation functions to maximize the model robustness. Using a meta-heuristic search method, SARAF can handle the intractable complexity of the original search problem Salimi et al. (2023b;a).

## 3 METHODOLOGY

From the concept of non-robust features, it is understood that through adversarial training, these features become less informative about the output label Ilyas et al. (2019). From an information-theoretic point-of-view, we also recognize that as the occurrence probability of a random vector increases, its informational value decreases Cover & Thomas (2006). Consequently, to identify the less informative features, it is useful to examine the regions of high probability within the feature space. One easily accessible high-probability point is the feature mean. Following this reasoning, we can block minor variations around the mean to eliminate the less informative or equivalently, the non-robust features. We begin this section by designing a regularized optimization objective to block non-robust features. Although this optimization problem is not used in the final MEANSPARSE technique, its parameter update rule provides valuable insight into the design of the sparsification operator introduced later. Finally, we demonstrate the complete MEANSPARSE technique.

### 3.1 INTUITION FROM REGULARIZED OPTIMIZATION OBJECTIVE

The small changes around the feature mean value at the output of different layers are potentially non-robust features. Thus we start by penalizing those changes using $\ell_0$ norm in an arbitrary layer $l$ for a training sample $\boldsymbol{x}$ (extension to the whole dataset and all layers is straightforward). The original training objective can be formulated as:

$$\mathcal{P}_0: \ \boldsymbol{\theta}_0^\star = \arg\min_{\boldsymbol{\theta}} \ L(\boldsymbol{\theta}) + \gamma\|\overline{\boldsymbol{a}}(\boldsymbol{x};\boldsymbol{\theta})\|_0, \ \ L(\boldsymbol{\theta}) \triangleq \mathcal{L}(\boldsymbol{y}(\boldsymbol{x}), \boldsymbol{y}^\star; \boldsymbol{\theta}) \tag{3}$$

where $\boldsymbol{\theta}$ is the model parameters vector, $\gamma$ is a hyper-parameter, $\overline{\boldsymbol{a}}(\boldsymbol{x}; \boldsymbol{\theta})$ is the centralized feature vector before the activation function in the regularized layer, $\boldsymbol{y}(\cdot)$ is the model output and $\boldsymbol{y}^{\star}$ is the target label corresponding to $\boldsymbol{x}$. Throughout the paper, we assume $\mu_a$ to be a fixed vector that is updated using an exponential filter during training, similar to the mean and variance vectors calculated in a batch normalization layer. Figure 2 represents the way mean-centered features for regularization are generated. The optimization problem in equation 3 can be solved via penalty method.

To solve problem equation 3, we introduce an approximate version of this problem based on the penalty method (for simplicity we use $\overline{\boldsymbol{a}}$ instead of $\overline{\boldsymbol{a}}(\boldsymbol{x}; \boldsymbol{\theta})$):

$$\mathcal{P}_\lambda : \ \boldsymbol{\theta}_{0,\lambda}^{\star}, \boldsymbol{w}^{\star} = \underset{\boldsymbol{\theta}, \boldsymbol{w}}{\arg\min} \ L(\boldsymbol{\theta}) + \gamma\|\boldsymbol{w}\|_0 + \frac{1}{2\lambda}\|\boldsymbol{w} - \overline{\boldsymbol{a}}\|_2^2 \tag{4}$$

where $\lambda$ is the penalty parameter, $\boldsymbol{w}$ is the penalty variable, and the optimization is with respect to both variables $\boldsymbol{\theta}$ and $\boldsymbol{w}$. Based on the penalty method (PM), we can conclude: $\lim_{\lambda \to 0} \boldsymbol{\theta}_{0,\lambda}^{\star} = \boldsymbol{\theta}_0^{\star}$. So to approximate $\boldsymbol{\theta}_0^{\star}$, one can solve problem $\mathcal{P}_\lambda$ and decrease $\lambda$ during the training. One approach to solve problem $\mathcal{P}_\lambda$ is to use block coordinate descent where the $\boldsymbol{\theta}$ and $\boldsymbol{w}$ are updated sequentially Nocedal & Wright (1999). So we have two steps in each training iteration $k$.

**Step1$\rightarrow$ Calculatign $\boldsymbol{w}_k$:** To calculate $\boldsymbol{w}_k$, we should solve:

$$\boldsymbol{w}_k = \underset{\boldsymbol{w}}{\arg\min} \ \gamma\lambda\|\boldsymbol{w}\|_0 + \frac{1}{2}\|\boldsymbol{w} - \overline{\boldsymbol{a}}_{k-1}\|_2^2 \tag{5}$$

The problem in equation 5 matches the definition of the proximal operator in equation 1 and can be solved exactly by the proximal operator for the function $f(\boldsymbol{w}) = \gamma\lambda\|\boldsymbol{w}\|_0$. Thus we have:

$$\boldsymbol{w}_k = \underset{f}{\mathrm{prox}}(\overline{\boldsymbol{a}}_{k-1}) = \mathcal{H}_{2\lambda\gamma}(\overline{\boldsymbol{a}}_{k-1}) \tag{6}$$

**Step 2 $\rightarrow$ Calculating $\boldsymbol{\theta}_k$:** To calculate $\boldsymbol{\theta}_k$, we have to solve the following problem:

$$\boldsymbol{\theta}_k = \underset{\boldsymbol{\theta}}{\arg\min} \ L(\boldsymbol{\theta}) + \frac{1}{2\lambda}\|\boldsymbol{w}_k - \overline{\boldsymbol{a}}\|_2^2 \tag{7}$$

while $\boldsymbol{w}_k$ is calculated using hard-thresholding operator as in equation 6. One step of (stochastic) gradient descent can be used to update $\boldsymbol{\theta}$ based on equation 7.

## 3.2 MEAN-CENTERED FEATURE SPARSIFICATION

Update rule equation 7 shows $\overline{\boldsymbol{a}}$ approaches $\mathcal{H}_{2\lambda\gamma}(\overline{\boldsymbol{a}}_{k-1})$ by the second term while the weight for this term is increasing (equivalently $\lambda$ is decreasing) during the training. Based on this intuition, we propose to use the $\mathcal{H}_{2\lambda\gamma}(\overline{\boldsymbol{a}}_{k-1})$ in the forward propagation of the model. If we add the element-wise mean subtraction and addition in 2 into the hard-thresholding operator, we achieve the curve in the bottom of the first column in 1 assuming $\alpha\sigma_{\mathrm{ch}} = (2\lambda\gamma)^2$ (for simplicity of notation, we use $Th$ instead). This curve represents a sparsification around the feature mean value $\mu_a$. In other words, we have the following element-wise operation over the input $\boldsymbol{a}^{(\mathrm{in})}$:

$$a^{(\mathrm{out})} = \begin{cases} \mu_a & if \ |a^{(\mathrm{out})} - \mu_a| \leq \ Th \\ a^{(\mathrm{in})} & if \ |a^{(\mathrm{out})} - \mu_a| > \ Th \end{cases}$$

So, features within the $Th$-vicinity of the feature mean are blocked and the mean value is output, while larger values pass through the Sparsification operator unchanged.

## 3.3 MEANSPARSE TECHNIQUE

Now we have a mean-centered sparsification operator that can be integrated into the model to block non-robust variations around the mean value. However, there are several challenges and improvements to address, which will be discussed in this section. By applying these enhancements, the final technique is formed.

### SPARSIFICATION IN POST-PROCESSING

As a general input-output mapping relation, mean-centered feature sparsification can be added to any part of deep learning architecture without changing the dimension. The back-propagation of training signals through this sparsification operator will be zero for inputs in the $Th$-vicinity of the feature mean, which avoids model training since the mean is the place where the highest rate of features lies. Two options can be applied to solve this problem.

The first approach is to gradually increase the $Th$ value through iterations. At the start of training, $Th$ is set to zero. Thus, sparsification reduces to an identity transformation, which solves the problem of backpropagation. Then, throughout the iterations, $Th$ is increased until it reaches a predefined maximum value. The most challenging problem in this scenario is the selection of a $Th$ scheduler as the model output is not differentiable with respect to $Th$ and $Th$ cannot be updated using gradient. In this scenario, we need to calculate the mean vector for the input feature representation inside the sparsification operator in a similar way to batch normalization layers.

The second solution is to apply sparsification as a post-training step to an adversarially trained model. In this scenario, first, we freeze the model weights and add the mean-centered feature sparsification operator to the model in the predefined positions. Then we need to pass through the complete training set once to calculate the feature mean value. Next, we decide on the value of $Th$. Similar to the previous scenario, this step cannot be done using gradient-based methods as the operator is not differentiable with respect to $Th$. Thus, we can search over a range of $Th$ values and find the best one that results in the highest robustness while maintaining the model's clean accuracy.

The two approaches differ significantly: while training with MeanSparse can improve robustness by influencing learned representations, it requires a carefully designed threshold scheduler to avoid issues like gradient zeroing or instability, particularly in large models, with early experiments showing unstable training and failed convergence. In contrast, the post-training approach integrates more easily, leveraging established model statistics and requiring only a search over alpha values, making it scalable to large models. It has been successfully applied to architectures like Swin-L, achieving a $+2.56\%$ robustness improvement without destabilization. Given the challenges of training MeanSparse in large models and the success of post-training integration, we chose the latter. This involves freezing the model, applying sparsification over mean-centered features before activation functions, calculating the mean value over the training set, and determining the optimal threshold $(Th)$ to maximize robustness while maintaining clean accuracy.

### ADAPTIVE SPARSIFICATION USING FEATURE STANDARD DEVIATION

As we need to apply sparsification before all activations in the model, searching over a simple space and finding a suitable set of $Th$ for different activation functions becomes intractable, even in small models. Thus, we design to select the threshold as $Th = \alpha \times \sigma_a$. While we pass through the training set to find the mean value $\mu_a$, we can also determine the variance value $\sigma_a^2$ in parallel and use $Th = \alpha \times \sigma_a$ as the threshold. As a result, for a fixed value of $\alpha$, the blocking vicinity around the mean increases as the variance of the feature increases.

### PER-CHANNEL SPARSIFICATION

Throughout the neural network, the input to the activation functions can be considered as 4-dimensional, represented as $N \times C \times H \times W$ ($N$, $C$, $H$ and $W$ represents the batch size, channels, height and width, respectively). To better capture the statistics in the representation, we use per-channel mean $\mu_{ch}$ and variance $\sigma_{ch}$ in sparsification operator. As a result, the mean and variance vectors are $C$-dimensional vectors. The sparsification operates on each feature channel separately, using the mean and variance of the corresponding channel.

### COMPLETE PIPELINE

The MEANSPARSE pipeline integrates mean-centered sparsification with three key modifications, placing it before all activation functions with a shared $\alpha$ parameter. Figure 1 shows an example applied to an adversarially trained RaWideResNet on the CIFAR-10 dataset. The first column illustrates a histogram of one channel's features in the first layer, alongside its mean $\mu_{ch}$ and standard

deviation $\sigma_{\text{ch}}$. The corresponding sparsification operator blocks variations near $\alpha\sigma_{\text{ch}}$ around the mean. The second and third columns show the input and output of this operator for a test image from the CIFAR-10 dataset in sample space and histogram. Notably, the feature histogram is bimodal, with one mode aligning with $\mu_{\text{ch}}$, representing uninformative variations blocked by the operator.

By using the MEANSPARSE presented in this paper, one can enhance the SOTA robust accuracy over CIFAR-10, CIFAR-100 Krizhevsky & Hinton (2009) and ImageNet Deng et al. (2009) datasets, as evidenced by the RobustBench Croce et al. (2020) rankings. Through various ablation studies, we also demonstrate the effectiveness of the proposed pipeline in multiple scenarios. This pipeline sheds light on a new approach to improve robustness without compromising clean accuracy.

## 4 EXPERIMENTS

The proposed pipeline in Figure 1 represents the way one can integrate MEANSPARSE into different trained models. In this section, we present several experiments to demonstrate the effectiveness of the MEANSPARSE across different architectures and datasets and in improving the SOTA robustness. The experiments were conducted using an NVIDIA A100 GPU.

### 4.1 EVALUATION METRICS

Throughout the experiments, we use *Clean* to represent the model accuracy for clean data. *A-PGD$_{ce}$* is used as a robustness measure and represents accuracy after applying Auto PGD with Cross-Entropy (CE) loss. *A-PGD* is used as another robustness measure and represents accuracy after applying Auto-PGD with both CE and Difference of Logits Ratio (DLR) loss metrics. Finally, we use *AA* to represent AutoAttack accuracy. All the robustness metrics can be found in Croce & Hein (2020). In our experiments, MEANSPARSE is applied post-training, eliminating the randomness typically introduced during training. Moreover, results from RobustBench for *A-PGD$_{ce}$*, *A-PGD*, and *AA* accuracies are highly consistent, showing negligible variability. Consequently, statistical significance tests are almost zero for all the results reported in this section.

### 4.2 INTEGRATION TO STATE-OF-THE-ART MODELS

With the introduction of adversarial examples for deep learning models, their reliability was severely questioned Carlini & Wagner (2017). However, several models have now been developed that present a high level of accuracy while also maintaining acceptable robustness against attacks imperceptible to humans Croce et al. (2020). In this part, we focus on the trained models with leading performance over CIFAR-10, CIFAR-100 Krizhevsky & Hinton (2009) and ImageNet Deng et al. (2009) datasets. We use the trained models and the proposed pipeline to integrate MEANSPARSE before all the activation functions. Then we evaluate the performance of the resulting models.

Figure 3 shows that integrating MEANSPARSE into the SOTA robust model in the RobustBench ranking Croce & Hein (2020) across various benchmark datasets achieves a new SOTA in robustness. Figure 3(a) depicts the results for the Clean and AA metrics of the standard and MEANSPARSE-integrated versions of the WideResNet-94-16 Bartoldson et al. (2024), RaWideResNet-70-16 Peng et al. (2023) and WideResNet-70-16 Wang et al. (2023) over CIFAR-10 dataset. The original WideResNet-94-16 Bartoldson et al. (2024) currently holds the SOTA performance in the RobustBench ranking Croce & Hein (2020) for $\ell_\infty$ untargeted attack. After post-processing with the proposed pipeline, the Clean accuracy drops slightly (from $93.68\%$ to $93.63\%$), while the AA significantly increases (from $73.71\%$ to $75.28\%$), setting a new record for robustness in terms of AA on the CIFAR-10 dataset. Similar results can be obtained for RaWideResNet-70-16 Peng et al. (2023) and WideResNet-70-16 Wang et al. (2023). Also, as a result of post-processing the robust model based on the WideResNet-70-16 architecture, the resulting AA accuracy ($71.41\%$) exceeds this value for the robust model based on the RaWideResNet architecture ($71.07\%$), even though WideResNet has a less complex architecture compared to RaWideResNet, which is equipped with the Squeeze-and-Excitation module.

Figure 3(b) represents the result of integrating MEANSPARSE into the current SOTA robust model (WideResNet-70-16 Wang et al. (2023)) in terms of AA accuracy with $\ell_2$ distance metric. Similarly, while the clean accuracy is almost unchanged (from $95.54\%$ to $95.49\%$), the AA accuracy is improved (from $84.97\%$ to $87.28\%$).

Figure 3(c) illustrates the clean and AA accuracy for three different models adversarially trained on the ImageNet dataset (Swin-L Liu et al. (2024), ConvNeXt-L Liu et al. (2023) and RaWideResNet Peng et al. (2023)). By carefully selecting the $\alpha$ value, while the Clean accuracy remains unchanged, the AA accuracy improves significantly. The Swin-L model Liu et al. (2024) is currently the SOTA robust model over the ImageNet dataset. When integrating this model with MEANSPARSE, while clean accuracy drops slightly (from 78.92% to 78.86%), the AA accuracy is increased (from 59.56% to 62.12%) setting a new SOTA.

Figure 3(d) shows the results of integrating MEANSPARSE with the current state-of-the-art (SOTA) robust model on the CIFAR-100 dataset (WideResNet-70-16 Wang et al. (2023)). For this model, the clean and AutoAttack (AA) accuracies are 75.22% and 42.67%, respectively. After integrating MEANSPARSE, these accuracies change to 75.17% and 44.78%, establishing a new SOTA robust model with nearly identical clean accuracy.

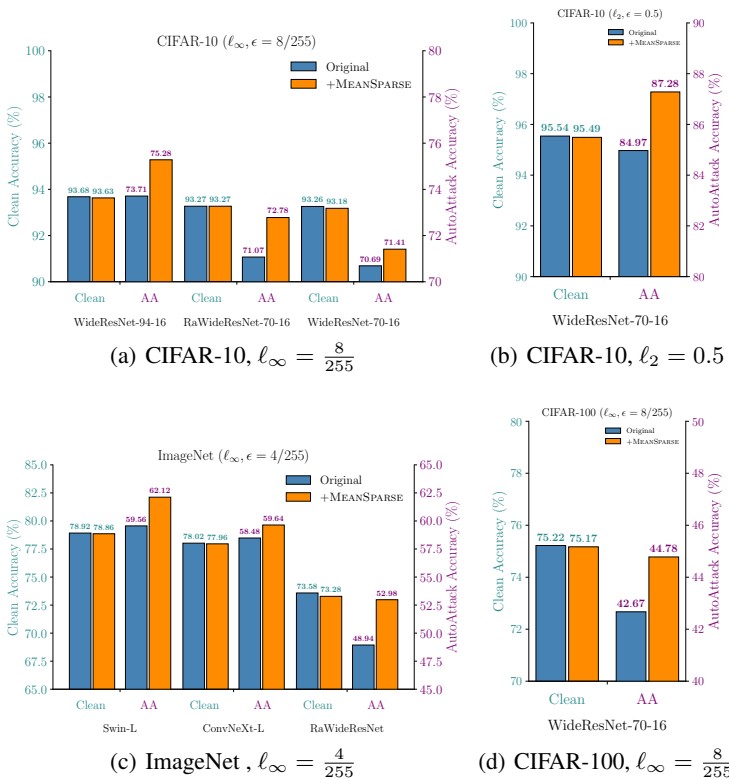

Figure 3: Original models performance along with their performance after integrating with MEANSPARSE technique. For CIFAR-10 dataset with $\ell_\infty$ attack, we have WideResNet-94-16 (Rank 1) Bartoldson et al. (2024), RaWideResNet-70-16 (Rank 4) Peng et al. (2023) and WideResNet-70-16 (Rank 5) Wang et al. (2023) while for $\ell_2$ attack we have WideResNet-70-16 (Rank 1) Wang et al. (2023). For ImageNet dataset with $\ell_\infty$ attack, we have Swin-L (Rank 2) Liu et al. (2024), ConvNeXt-L (Rank 4) Liu et al. (2023) and RaWideResNet (Rank 12) Peng et al. (2023). For CIFAR-100 dataset with $\ell_\infty$ attack, we have WideResNet-70-16 (Rank 1) Wang et al. (2023) (all the rankings are based on RobustBench Croce et al. (2020))

Table 1 presents the results of integrating MEANSPARSE into RaWideResNet Peng et al. (2023) and WideResNet Wang et al. (2023) architectures trained on the CIFAR-10 dataset and ConvNeXt-L Liu et al. (2023) and RaWideResNet Peng et al. (2023) on the ImageNet dataset using adversarial training. The results are provided for different values of $\alpha$ (note that the threshold equals $\alpha \times \sigma_{ch}$). The experiment results over the CIFAR-10 test set depict that for a range of $\alpha$ below 0.25, the clean accuracy has decreased negligibly or almost unchanged while APGD$_{ce}$ is increasing. Although these results may be attributed to overfitting, the result over the CIFAR-10 training set also represents a similar trend. So an important takeaway from CIFAR-10 results in Table 1 is that the variations

around the mean value of representation can be considered as a capacity provided by the model for the attackers while its utilization is limited. MEANSPARSE requires one pass over CIFAR-10 training set to calculate the statistics. The time for this pass is 102 seconds and 95 seconds for the RaWideResNet and WideResNet architectures, respectively. The increase in $APGD_{ce}$ evaluation time after MEANSPARSE integration is negligible, with only a $2\%$ increase in time.

If this capacity is controlled suitably, then the robustness of the model will be increased (model capacity for attacker will be decreased) while model utilization will be unchanged. So by increasing $\alpha$ from zero, we decrease the attacker capacity while user utilization is unchanged. By increasing $\alpha$ to larger values, we start to decrease user utilization, too. Although there is no rigid margin between these two cases, one can find suitable $\alpha$ values that limit the attacker side while user utilization is unchanged.

Results over ImageNet presented in Table 1 show that similar outcomes are achieved when MEANSPARSE is integrated into models based on the ConvNeXt-L Liu et al. (2023) and RaWideResNet Peng et al. (2023) architectures adversarially trained. Note that throughout our experiment over the RaWideResNet Peng et al. (2023) architecture for both CIFAR-10 and ImageNet-1k datasets, we do not apply the sparsification operator inside the Squeeze-and-Excitation module. The time for the MEANSPARSE pass over the ImageNet-1K training set is 89 minutes for the ConvNeXt-L architecture and 101 minutes for the RaWideResNet architecture. The increase in $APGD_{ce}$ evaluation time after MEANSPARSE integration is also negligible, with only a $3\%$ increase in time.

Table 1: Clean accuracy and accuracy under A-PGD-CE attack Croce & Hein (2020) (represented by $APGD_{ce}$) for leading robust models on the CIFAR-10 and ImageNet-1k datasets (*Base* row) and after their integration with the MEANSPARSE technique (RaWideResNet Peng et al. (2023) and WideResNet Wang et al. (2023) are leading models in Robustbench ranking for CIFAR-10 dataset and ConvNeXt-L Liu et al. (2023) and RaWideResNet Peng et al. (2023) are highly ranked models for ImageNet dataset in Robustbench Croce et al. (2020)). The best result in each column is in **bold** and the threshold selected through the proposed pipeline is in blue.

| | CIFAR-10 | | | | | | | | ImageNet-1k | | | |
| | RaWideResNet | | | | WideResNet | | | | ConvNeXt-L | | RaWideResNet | |
| | Train | | Test | | Train | | Test | | Test | | Test | |
| $\alpha$ | Clean | $APGD_{ce}$ | Clean | $APGD_{ce}$ | Clean | $APGD_{ce}$ | Clean | $APGD_{ce}$ | Clean | $APGD_{ce}$ | Clean | $APGD_{ce}$ |
|---|---|---|---|---|---|---|---|---|---|---|---|---|
| Base | **99.15** | 93.44 | **93.27** | 73.87 | **99.37** | 94.69 | **93.26** | 73.45 | 78.02 | 60.28 | **73.58** | 52.24 |
| 0.15 | **99.15** | 93.69 | **93.27** | 74.40 | 99.36 | **94.94** | 93.15 | 73.92 | 78.00 | 60.62 | 73.24 | 53.38 |
| 0.20 | 99.13 | **93.75** | 93.26 | 74.52 | 99.33 | 94.93 | 93.18 | 74.11 | **78.04** | 60.88 | 73.28 | 53.78 |
| 0.25 | 99.10 | 93.71 | 93.24 | **74.74** | 99.32 | 94.85 | 93.03 | **74.12** | 77.94 | 61.12 | 72.82 | **53.92** |
| 0.30 | 99.07 | 93.49 | 93.17 | 74.63 | 99.29 | 94.58 | 92.90 | 74.04 | 77.96 | 61.48 | 72.36 | 53.86 |
| 0.35 | 98.98 | 93.12 | 93.04 | 74.55 | 99.23 | 94.07 | 92.75 | 73.72 | 77.74 | 61.54 | 71.74 | 52.92 |
| 0.40 | 98.87 | 92.48 | 92.88 | 73.90 | 99.11 | 93.27 | 92.38 | 73.32 | 77.56 | 61.84 | 69.90 | 51.68 |
| 0.45 | 98.70 | 91.37 | 92.60 | 73.36 | 98.92 | 91.93 | 92.08 | 72.27 | 77.12 | 62.18 | 67.16 | 48.74 |
| 0.50 | 98.42 | 89.76 | 92.37 | 72.37 | 98.66 | 90.17 | 91.55 | 71.10 | 76.90 | **62.32** | 61.78 | 43.06 |

## 4.3 INSIGHTS FROM ABLATION STUDY

We conducted several experiments to explore different aspects of MEANSPARSE integration into trained models. These experiments examine the effects of activation functions A.1, the type of adversarial attack (either $\ell_\infty$ or $\ell_2$) A.2, the robustness of MEANSPARSE integrated models against black-box attacks A.3, the effect of adaptive attack with access to MEANSPARSE modules location and specifications (mean and variance vectors) A.4, the types of adversarial examples used during training with AT A.5, the effect of integrating a standard trained model with MEANSPARSE A.6, the impact of centering MEANSPARSE around the mean value A.7, the influence of attack power on MEANSPARSE integration performance A.8, the effects of integrating MEANSPARSE with different patterns, such as isolated integration with a single activation A.9, and the robustness improvement resulting from MEANSPARSE integration in terms of PGD-50 attack A.10. The main takeaways from our ablation study can be summarized as:

- MEANSPARSE effectively enhances robustness across various activation functions. In other words, the activation function alone does not effectively eliminate non-robust features, which are previously reduced during training.
- By calculating the sparsity using other well-known adversarial attacks such as PGD-50 Madry et al. (2018), we observe a similar improvement in robustness, consistent with the AA and APGD accuracy values.
- Although MEANSPARSE masks the gradient, which can explain the robustness improvement against white-box attacks, it also demonstrates enhanced robustness when evaluated using black-box attacks.
- A straightforward attack aiming to minimize the impact of MEANSPARSE in affected regions is generally ineffective at significantly compromising the robustness of MEANSPARSE-integrated models. However, as noted in the limitations, alternative strategies, such as treating MEANSPARSE as an identity function during backpropagation, can reduce the robustness provided by MEANSPARSE (see Section A.4 for details on adaptive attacks).
- MEANSPARSE is effective against both $\ell_\infty$ and $\ell_2$ attacks. Specifically, if a model is fortified against $\ell_\infty$ attacks using AT, adding MEANSPARSE enhances robustness against both $\ell_\infty$ and $\ell_2$ attacks. Also, MEANSPARSE is not sensitive to the method used to generate adversarial samples during AT.
- Using dedicated mean and variance for each channel in the MEANSPARSE technique is critical to its performance. Otherwise, we cannot expect the observed improvement in robustness.
- Integrating MEANSPARSE in isolated activation functions remains effective for robustification, although the improvement is limited.

### 4.4 LIMITATIONS

The MEANSPARSE technique proposed in this paper is a post-training method that is useful on top of adversarially trained models. Using MEANSPARSE for models trained in a standard way does not lead to robustness improvement as we expect based on our intuition introduced in this paper. Additionally, while integrating a model with MEANSPARSE shows clear improvements against both white-box and black-box non-adaptive attacks, white-box attacks that ignore the MeanSparse transformation during backpropagation and use the identity function can impact MEANSPARSE's effectiveness.

## 5 CONCLUSIONS

Adversarial training, a widely accepted defense against evasion attacks, attenuates non-robust feature extractors in a model. Although adversarial training (AT) is effective, this paper demonstrates that there is still an easily accessible capacity for attackers to exploit. We introduce the MEANSPARSE technique, which partially blocks this capacity in the model available to attackers and enhances adversarial robustness. This technique, easily integrable after training, improves robustness without compromising clean accuracy. By integrating this technique into trained models, we achieve a new record in adversarial robustness on CIFAR-10 for both $\ell_\infty$ and $\ell_2$ untargeted attacks, as well as on CIFAR-100 and ImageNet for $\ell_\infty$ untargeted attacks.

### ETHICAL CONSIDERATIONS

This research upholds rigorous ethical standards, adhering to institutional and international guidelines. The societal implications were critically evaluated, balancing potential benefits such as enhanced model robustness and trust in AI against risks like misuse and bias reinforcement. Mitigation strategies, including controlled dissemination, fairness audits, and transparency measures, were proposed to address these risks. Transparency and accountability were central to the study, with no conflicts of interest impacting its outcomes.

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

# A ABLATION EXPERIMENT

We conducted several experiments to evaluate the effectiveness of MEANSPARSE in various scenarios. In the majority of experiments, we use APGD accuracy (accuracy after attacking the model with Auto-PGD using both Cross-entropy and Difference of Logits Ratio loss metrics) as the robustness metric Croce & Hein (2020). During our simulations, we found APGD accuracy to be almost equal to AA accuracy, with the advantage that APGD can be measured faster.

## A.1 ACTIVATION FUNCTION

MEANSPARSE is generally appended before the activations function. This questions the effectiveness of MEANSPARSE for different activation functions. In this experiment, we check the performance of MEANSPARSE when integrating the models with different activation functions. The specifications of this experiment are:

- Dataset: CIFAR-10 Krizhevsky & Hinton (2009)

- Architecture: ResNet-18 He et al. (2015)

- Optimizer: SGD (initial learning rate: 0.1, weight decay: 0.0005, momentum: 0.9)

- Number of Epochs: 200

- Batch Size: 256

- Learning rate scheduler: The initial learning rate of 0.1 is reduced by a factor of 10 at epoch 100 and 150.

- Best model selection: We evaluate the model at each epoch of training and select the one with the highest PGD adversarial accuracy on the test set.

- Adversarial training properties: 10 step PGD adversarial training Madry et al. (2018) with respect to $\ell_\infty$ attacks with radius $8/255$ and step size of 0.0078.

The experiments were conducted using an NVIDIA A100 GPU. Training each ResNet-18 model required approximately 6 hours of computational time on a single A100 GPU. In addition, the evaluation of each trained model took around 20 minutes on the same GPU.

Table 2 demonstrates the Clean and APGD accuracy metrics for a base ResNet-18 model, denoted by *Base*, with different activation functions and the same model post-processed by the proposed pipeline to integrate MeanSaprse. We use both non-parametric and parametric activation functions, such as PSSiLU Dai et al. (2022). In our experiment, the GELU activation function, proposed in Hendrycks & Gimpel (2018) and generally used in attention-based architectures, results in the best APGD accuracy. When this model is integrated with MEANSPARSE, we observe a significant improvement in APGD accuracy ($49.12\% \rightarrow 50.37\%$) while the clean accuracy slightly decreases ($84.59\% \rightarrow 84.32\%$) for $\alpha = 0.2$. Additionally, if we can tolerate $\sim 1\%$ reduction in clean accuracy, we can increase APGD accuracy by more than $2\%$. The first important takeaway is the effectiveness of MEANSPARSE integration with different activations and the inability of the activation function to replicate the role of the MEANSPARSE technique. The second is that $\alpha = 0.2$ consistently leads to large improvements in APDG accuracy with a negligible reduction in clean accuracy for all activation functions. Even for the ReLU activation function, $\alpha = 0.2$ increases both Clean and APGD accuracies simultaneously. We will use the ResNet-18 model trained adversarially with GELU activation over the CIFAR-10 dataset for our future experiments.

## A.2 THREAT MODEL

In this experiment, we focused on the effectiveness of MEANSPARSE against two well-studied threat models, $\ell_\infty$ and $\ell_2$. Until now, our focus was on the $\ell_\infty$ threat model, which generally results in lower robust accuracy Croce et al. (2020). On the other hand, $\ell_2$ threat models can focus on a specific part of the image, and this localization of adversarial perturbation may challenge the effectiveness of the MEANSPARSE technique. In this experiment, we compare the robustness against both $\ell_\infty$ and $\ell_2$ threat models. The model is ResNet-18 with the GELU activation function as in our *Activation Function* experiment in 2. Table 3 illustrates the clean and APGD accuracy for $\ell_\infty$ and $\ell_2$ threats. As the accuracy for the $\ell_\infty$ threat model increases with increasing $\alpha$, the accuracy for $\ell_2$ also increases,

Table 2: Clean and APGD accuracy of ResNet-18 model with different activation functions over CIFAR-10 test set before (Base) and after integration with MEANSPARSE technique for different value of $\alpha$

| | Activation | | | | | | | | | | | |
| --- | --- | --- | --- | --- | --- | --- | --- | --- | --- | --- | --- | --- |
| | ReLU | | ELU | | GELU | | SiLU | | PSiLU | | PSSiLU | |
| $\alpha$ | Clean | APGD | Clean | APGD | Clean | APGD | Clean | APGD | Clean | APGD | Clean | APGD |
| Base | 83.59 | 47.77 | **81.65** | 46.65 | 84.59 | 49.12 | 83.05 | 48.54 | **84.80** | 48.02 | 83.95 | 48.03 |
| 0.05 | 83.58 | 47.89 | 81.64 | 47.47 | **84.63** | 49.25 | 83.10 | 48.70 | **84.80** | 48.11 | **83.97** | 48.22 |
| 0.1 | 83.59 | 48.25 | 81.58 | 48.69 | 84.58 | 49.57 | 83.09 | 49.05 | 84.72 | 48.58 | 83.95 | 48.61 |
| 0.15 | 83.57 | 48.60 | 81.53 | 49.22 | 84.42 | 50.00 | **83.11** | 49.73 | 84.66 | 48.88 | 83.93 | 49.05 |
| 0.2 | **83.71** | 48.88 | 81.53 | **49.49** | 84.32 | 50.37 | 83.01 | 50.01 | 84.69 | 49.07 | 83.79 | 49.29 |
| 0.25 | 83.46 | **49.30** | 81.56 | 49.14 | 84.18 | 50.80 | 83.00 | 50.59 | 84.61 | **49.36** | 83.75 | **49.32** |
| 0.3 | 83.48 | 49.24 | 81.20 | 48.66 | 83.95 | 51.13 | 82.78 | **50.81** | 84.27 | 49.35 | 83.81 | 49.19 |
| 0.35 | 83.13 | 49.27 | 80.34 | 47.60 | 83.51 | **51.36** | 82.49 | 50.79 | 83.76 | 49.25 | 83.55 | 49.11 |
| 0.4 | 82.49 | 48.87 | 78.77 | 46.10 | 82.85 | 51.25 | 81.34 | 49.97 | 83.20 | 48.76 | 82.69 | 48.63 |

which depicts the effectiveness and generalization of MEANSPARSE across well-studied threat models. Additionally, the improvement compared to the base model at $\alpha = 0.2$ is larger for the $\ell_2$ threat than for the $\ell_\infty$ threat.

Table 3: Clean and APGD accuracy of ResNet-18 model with GELU activation function over CIFAR-10 test set before (Base) and after integration with MEANSPARSE technique for different value of $\alpha$ and different threat models from RobustBesnch Croce & Hein (2020)

| | Threat Model | | |
| --- | --- | --- | --- |
| | | $\ell_\infty, \epsilon = \frac{8}{255}$ | $\ell_2, \epsilon = \frac{128}{255}$ |
| $\alpha$ | Clean | APGD | APGD |
| Base | 84.59 | 49.12 | 59.98 |
| 0.05 | **84.63** | 49.25 | 60.23 |
| 0.10 | 84.58 | 49.57 | 60.53 |
| 0.15 | 84.42 | 50.00 | 60.87 |
| 0.20 | 84.32 | 50.37 | 61.36 |
| 0.25 | 84.18 | 50.80 | 61.60 |
| 0.30 | 83.95 | 51.13 | 61.73 |
| 0.35 | 83.51 | **51.36** | **61.74** |
| 0.40 | 82.85 | 51.25 | **61.74** |

## A.3 BLACKBOX ATTACKS

The MEANSPARSE operation masks the gradient during backward propagation when the input $a^{\text{in}}$ lies within $\alpha\sigma_{\text{ch}}$ of the channel mean $\mu_{\text{ch}}$ (see the bottom plot in the first column of Figure 1). An interesting direction to explore is the performance of black-box attacks that do not rely on gradients, both before and after integrating MEANSPARSE. For this experiment, we select the Square Attack Andriushchenko et al. (2020). Table 4 shows the clean and robust accuracy for a ResNet-18 model with GELU activation before (Base) and after integration with MEANSPARSE (robust accuracy is denoted by *Square*, representing the accuracy under Square Attack). The results indicate a similar improvement in robust accuracy on par with APGD, while clean accuracy remains largely unaffected across a wide range of $\alpha$ values, from 0.05 to 0.2.

Table 4: Clean and Square accuracy (Square accuracy in the *Square* column represents the accuracy after applying square attack Andriushchenko et al. (2020)) of ResNet-18 model with GELU activation function over CIFAR-10 test set before (Base) and after integration with MEANSPARSE technique for different value of $\alpha$

| $\alpha$ | Clean | Square |
|------|-------|--------|
| Base | 84.59 | 57.37 |
| 0.05 | **84.63** | 57.66 |
| 0.10 | 84.58 | 58.19 |
| 0.15 | 84.42 | 59.31 |
| 0.20 | 84.32 | 60.30 |
| 0.25 | 84.18 | 60.67 |
| 0.30 | 83.95 | 61.32 |
| 0.35 | 83.51 | **61.52** |
| 0.40 | 82.85 | 61.49 |

## A.4 ADAPTIVE ATTACK

In this experiment, we investigate the potential for the attacker to enhance the attack by utilizing the knowledge that MEANSPARSE is employed to robustify the model. We propose a formulation to adapt the original PGD attack, incorporating the integration of MEANSPARSE to create a more powerful attack. In this scenario, we assume the attacker has complete access to the locations, mean, and variance vectors of the integrated MEANSPARSE operator throughout the model. Let $x$ represent the input image and $\{z_i\}$ the corresponding representations at the input of the MEANSPARSE modules in the model. When a perturbation $\delta$ is added to the input $(x + \delta)$, it results in new representations $\{z_i^\delta\}$. In the adaptive PGD attack, we solve the following optimization problem to determine the perturbation $\delta$:

$$\arg\max_{\delta} \quad \text{CE}(f(x + \delta), y) - \lambda \sum_i \|m_i \odot (z_i - z_i^\delta)\|_p^q$$

$$\text{subject to} \quad \|\delta\|_\infty \leq \epsilon, \quad x + \delta \in [0, 1]^D \tag{8}$$

where $\text{CE}(\cdot, \cdot)$ is the cross-entropy loss, $f$ is the model, $y$ is the true label, $\lambda$ is a hyper-parameter, $i$ indexes different MEANSPARSE modules in the model, $\odot$ is the element-wise multiplication operator, $\epsilon$ controls the perturbation imperceptibility via the $\ell_\infty$ norm, and $D$ is the dimension of the input image. $m_i$ represents a mask with the same dimensions as $z_i$, identifying the locations in $z_i$ affected by the MEANSPARSE module. For a given $z$, the corresponding $m$ can be computed element-wise as:

$$m_j = \begin{cases} 1 & \text{if } \mu_{\text{ch}} - \alpha\sigma_{\text{ch}} \leq z_j \leq \mu_{\text{ch}} + \alpha\sigma_{\text{ch}} \\ 0 & \text{Otherwise} \end{cases}, j = 1, 2, \ldots, K$$

where $\mu_{\text{ch}}$ and $\sigma_{\text{ch}}$ are the statistics of the MEANSPARSE module, and $K$ represents the representation dimension. Thus, whenever a feature in the input of MEANSPARSE is affected by this module for the input image $x$, the corresponding mask value becomes 1; otherwise, it is 0. As a result, the second term in the objective function of the adaptive PGD attack equation 8 penalizes perturbations that lead to changes in the features affected by MEANSPARSE. In the ideal case, the perturbation is designed such that none of the MEANSPARSE modules can block its effect in their input (we know that the changes in the input of the $i$-th MEANSPARSE module are $z_i - z_i^d$). Thus, the attacker can bypass the MEANSPARSE, which we denote as the Adaptive-$(p, q)$ attack.

To test the adaptive attack, we solve the optimization problem equation 8 using gradient ascent with a step size of $\mu = \frac{1}{255}$, a number of iterations $N = 50$, and $\epsilon = \frac{8}{255}$. When $\lambda = 0$, problem equation 8 reduces to the well-known PGD-50 attack, while for larger values of $\lambda$, we obtain an adaptive attack. We conducted experiments using this adaptive attack on a ResNet-18 model with a GELU activation function, which was adversarially trained on the CIFAR-10 dataset. Table 5 compares the results

of the PGD-50 attack with the Adaptive-$(1, 1)$ and Adaptive-$(2, 2)$ attacks before (Base) and after integration ($\alpha = 0.20$ and $\alpha = 0.35$) with MEANSPARSE. For the base model, all the attacks are equivalent, as $\boldsymbol{m}_i$ is a zero matrix for all MEANSPARSE modules. As $\alpha$ increases, the accuracy after applying our adaptive attack decreases compared to the PGD-50 attack, demonstrating the effectiveness of our adaptation. While the adapted attack is more powerful than the original PGD-50 attack, the improvement gap in robustness resulting from MEANSPARSE is still significant. For the base model, the robust accuracy is $51.56\%$, while the worst accuracies are $53.56\%$ and $55.26\%$ for $\alpha = 0.20$ and $\alpha = 0.35$, respectively, indicating a clear improvement in robustness, while the clean accuracy remains almost unchanged.

Table 5: Clean, PGD-50 and Adaptive Attack accuracy for a ResNet-18 model with GELU activation over CIFAR-10 dataset before (Base) and after integration with MEANSPARSE

| $\alpha$ | Clean | PGD-50 | Adaptive-$(1, 1)$ | Adaptive-$(2, 2)$ |
|---|---|---|---|---|
| Base | **84.59** | 51.56 | 51.56 | 51.56 |
| 0.20 | 84.32 | 53.89 | 53.56 | 53.58 |
| 0.35 | 83.51 | **55.50** | **55.36** | **55.26** |

## A.5 ADVERSARIAL TRAINING

MEANSPARSE is designed to integrate with models trained using adversarial training. In previous experiments, PGD Madry et al. (2018) was used for adversarial training. In this section, we compare the PGD results with those of TRADES Zhang et al. (2019) for adversarial training. For TRADES adversarial training, we set $\beta = 0.6$. For the other properties of adversarial training, we adopt the parameters specified in Section 2. Table 6 compares the results for base adversarially trained models and models integrated with MEANSPARSE. For TRADES, MEANSPARSE can still improve the APGD accuracy while maintaining clean accuracy. This table demonstrates overlapping information gathered around the mean feature values for both PGD and TRADES as adversarial training methods. In both methods, the information around the feature mean with a value of $\alpha \simeq 0.2$ is almost uninformative for clean data but can be utilized by attackers. Thus, the performance of MEANSPARSE represents generalizability across both PGD and TRADES adversarial training methods.

Table 6: Clean and APGD accuracy of ResNet-18 model with GELU activation function over CIFAR-10 test set before (Base) and after integration with MEANSPARSE technique for different value of $\alpha$ and adversarial training based on PGD Madry et al. (2018) and TRADES Zhang et al. (2019)

| | Adversarial training | | | |
|---|---|---|---|---|
| | PGD | | TRADES | |
| $\alpha$ | Clean | APGD | Clean | APGD |
| Base | 84.59 | 49.12 | 83.01 | 47.03 |
| 0.05 | **84.63** | 49.25 | 83.04 | 47.16 |
| 0.10 | 84.58 | 49.57 | 83.09 | 47.61 |
| 0.15 | 84.42 | 50.00 | **83.11** | 48.08 |
| 0.20 | 84.32 | 50.37 | 83.08 | 48.64 |
| 0.25 | 84.18 | 50.80 | 83.02 | 48.92 |
| 0.30 | 83.95 | 51.13 | 82.92 | **49.34** |
| 0.35 | 83.51 | **51.36** | 82.62 | 49.24 |
| 0.40 | 82.85 | 51.25 | 82.20 | 48.76 |

## A.6 STANDARD TRAINING

In the previous experiments, we explore the effect of two different adversarial training methods on performance improvement by MEANSPARSE. In this experiment, we test MEANSPARSE integration over a standard trained model. Table 7 represents the APGD accuracy over a ResNet-18 model with standard training for $\epsilon$ values starting from $1/255$ to $8/255$. The first column ($\alpha = 0$) is the base model performance while the next two columns ($\alpha = 0.2$ and $\alpha = 0.35$) represent the models integrated with MEANSPARSE. In all cases, the attacker can fool the model with imperceptible perturbation ($\epsilon \leq 4/255$) which necessitates the integration of MEANSPARSE with adversarially trained models.

Table 7: APGD accuracy of standard trained ResNet-18 model with GELU activation function over CIFAR-10 test set before (Base denoted by $\alpha = 0$ in the table) and after the integration of MEANSPARSE technique for different attack power $\epsilon$ and two different $\alpha$ values (0.20 and 0.35). Clean accuracy is provided next to each $\alpha$ value.

| | $\alpha$ | | |
|---|---|---|---|
| | 0 (94.36) | 0.2 (94.20) | 0.35 (93.61) |
| $\epsilon$ | APGD | APGD | APGD |
| $1/255$ | 49.09 | 52.06 | 56.36 |
| $2/255$ | 4.86 | 7.69 | 14.52 |
| $3/255$ | 0.07 | 0.24 | 1.33 |
| $4/255$ | 0.0 | 0.01 | 0.03 |
| $5/255$ | 0.0 | 0.0 | 0.01 |
| $6/255$ | 0.0 | 0.0 | 0.01 |
| $7/255$ | 0.0 | 0.0 | 0.01 |
| $8/255$ | 0.0 | 0.0 | 0.01 |

## A.7 FEATURE CENTERIZATION TYPE

The MEANSPARSE technique applies sparsification over mean-centered features across each channel in the input tensor to the activation function. In this section, we inspect the importance of the reference for feature centralization before sparsification in our proposed technique. For this purpose, we compare three cases where the reference for centering features is the channel mean ($\boldsymbol{\mu}_{ch}$), zero, and the mean over all features in all channels ($\mu$). For a fair comparison across all cases, we select the sparsification threshold equal to $\alpha \times \sigma_{ch}$, where $\sigma_{ch}$ emphasizes the channel-wise calculation standard deviation. Table 8 represents the result for different options of reference. When using zero as the reference for feature centralization, the improvement in APGD is higher than using the mean as the reference, but on the other hand, the utilization of the model also decreases. One challenging problem is the high rate of clean accuracy reduction when using zero as the reference, which may cause a huge reduction if not selected properly. When using the mean of all channels as the reference, the results in terms of both clean and APGD accuracy metrics underperform compared to the case where the channel mean is used as the reference. The main takeaway from the table is the importance of using channel-wise statistics to maintain utility while improving robustness.

## A.8 ATTACK POWER

In the previous experiments, the attack power was set to a fixed value of $8/255$ for $\ell_\infty$ attack threat which is generally the threshold of imperceptibility Croce & Hein (2020). In this experiment, we explore the effectiveness of the MEANSPARSE technique for a wide range of attack powers. Table 9 represents the APGD accuracy for $\epsilon$ values starting from $1/255$ to $16/255$. For $\alpha = 0.2$, the accuracy is smaller than the base model ($\alpha = 0$) for $1/255$ and $2/255$ values of $\epsilon$, which is generally due to the reduction of clean accuracy after applying MEANSPARSE (clean accuracy reduces to $84.32$ from $84.59$). For $\epsilon$ values larger than $2/255$, applying the MEANSPARSE technique persistently improves

Table 8: Clean and APGD accuracy of ResNet-18 model with GELU activation function over CIFAR-10 test set before (Base) and after the application of MEANSPARSE technique for different references for feature centralization

| | Reference for Centralization | | | | | |
| | zero | | $\mu$ | | $\boldsymbol{\mu}_{ch}$ | |
| $\alpha$ | Clean | APGD | Clean | APGD | Clean | APGD |
|---|---|---|---|---|---|---|
| *Base* | **84.59** | 49.12 | **84.59** | 49.12 | 84.59 | 49.12 |
| 0.05 | 84.58 | 49.32 | 84.58 | 49.22 | **84.63** | 49.25 |
| 0.1 | 84.50 | 49.63 | 84.52 | 49.45 | 84.58 | 49.57 |
| 0.15 | 84.30 | 50.26 | 84.36 | 49.79 | 84.42 | 50.00 |
| 0.2 | 84.06 | 50.98 | 84.13 | 50.23 | 84.32 | 50.37 |
| 0.25 | 83.55 | 51.58 | 83.83 | 50.73 | 84.18 | 50.80 |
| 0.3 | 82.81 | **52.08** | 83.27 | **50.87** | 83.95 | 51.13 |
| 0.35 | 81.68 | 51.95 | 82.52 | 50.61 | 83.51 | **51.36** |
| 0.4 | 79.84 | 51.72 | 81.09 | 49.93 | 82.85 | 51.25 |

the APGD accuracy. The same trend is also observed for $\alpha = 0.35$ except for the fact that after $\epsilon = 4/255$, we have persistent improvements over the base model.

Table 9: APGD accuracy of ResNet-18 model with GELU activation function over CIFAR-10 test set before (Base denoted by $\alpha = 0$ in the table) and after the application of MEANSPARSE technique for different attack power $\epsilon$ and two different $\alpha$ values (0.20 and 0.35). Clean accuracy is provided next to each $\alpha$ value.

| | $\alpha$ | | |
| | 0 (84.59) | 0.2 (84.32) | 0.35 (83.51) |
| $\epsilon$ | APGD | APGD | APGD |
|---|---|---|---|
| 1/255 | 81.10 | 80.97 | 79.60 |
| 2/255 | 77.72 | 77.60 | 76.47 |
| 3/255 | 73.37 | 73.54 | 72.64 |
| 4/255 | 68.88 | 69.08 | 68.80 |
| 5/255 | 64.31 | 64.70 | 64.48 |
| 6/255 | 58.95 | 59.92 | 60.15 |
| 7/255 | 54.11 | 55.15 | 55.71 |
| 8/255 | 49.12 | 50.37 | 51.36 |
| 9/255 | 42.77 | 44.41 | 45.72 |
| 10/255 | 37.48 | 39.17 | 40.77 |
| 11/255 | 32.62 | 34.20 | 36.04 |
| 12/255 | 28.12 | 29.77 | 31.50 |
| 13/255 | 23.77 | 25.44 | 27.18 |
| 14/255 | 19.42 | 20.76 | 22.77 |
| 15/255 | 15.51 | 17.01 | 19.09 |
| 16/255 | 11.97 | 13.41 | 15.44 |

## A.9 Applying to Specific Activation Functions

The MEANSPARSE technique is designed to sparsify mean-centered inputs to all activation functions in a model. In this experiment, we will focus on applying sparsification over a selected subset of activation functions. For this purpose, we index the activation functions in the ResNet-18 model. This model has 8 residual blocks, each with 2 activation functions: one in the *Main Path* and one after the result of the main path is added to the residual connection, totaling 16 activation functions. Additionally, this architecture has one activation function before the residual block. So, in total, we have 17 indices. Index 0 is the initial activation function. The odd indices are related to the activation function in the main path, and the even indices correspond to activation functions after the addition of the residual connection to the main path output. We select two scenarios. In the first scenario, denoted by *Single Activation*, we present the results when MEANSPARSE is applied to only the corresponding row-indexed activation function, while *Cumulative* depicts the result when MEANSPARSE is applied to the activation functions with indices less than or equal to the index represented in each row.

Table 10 represents the results for both *Single Activation* and *Cumulative* scenarios. One shared remark is that the starting layers are more effective in improving robustness. We can attribute this to the abstraction level in different layers of the ResNet-18 model. In the case of Single Activation, when we apply the MEANSPARSE technique to larger indices, we observe that not only does the robust accuracy decrease, but the clean accuracy increases. This indicates that for even better performance, we can use different $\alpha$ values for activation functions based on their depth level. Note that for the current version of the MEANSPARSE technique, we use a shared $\alpha$ for all activation functions.

Another aspect worth inspecting in this experiment is how the application of the MEANSPARSE technique to activation functions in the main path and after the addition of residual data to the main path result affects robustness in terms of APGD accuracy. So we consider two cases in Table 11. In the first case, MEANSPARSE is applied to all the activation functions in the main path of the ResNet block, denoted by *Main Path*. In the second case, MEANSPARSE is applied to all the activation functions after the addition of residual data to the output of the main path, denoted by *After Addition*. This table reveals an important result. While the application of MEANSPARSE to both types of activation functions contributes to the robustness of the model, the application of MEANSPARSE to the main flow of the model (*After Addition*) is more responsible for the improvements in the ranges where clean accuracy is almost unchanged ($\alpha \simeq 0.2$).

## A.10 Attack Type

Figure 3 demonstrates the improvement in robustness of leading models in RobustBench in terms of AutoAttack accuracy. In this experiment, we use PGD-50 Madry et al. (2018) for robustness measurement. Table 12 represents the Clean and PGD-50 accuracy values for different models before (denoted by Base) and after integration with MEANSPARSE (denoted by MEANSPARSE) over CIFAR-10, CIFAR-100 and ImageNet datasets. The results again confirm the improvements in Figure 3 as t integration with MEANSPARSE increases the PGD-50 accuracy in all models.

Table 10: Clean and APGD accuracy of ResNet-18 model with GELU activation function over CIFAR-10 test set after the application of MEANSPARSE technique for *Single Activation* and *Cumulative* cases. Single Activation represents the results when MEANSPARSE is applied to only the index activation function, and Cumulative Activation represents the results when MEANSPARSE is applied to the index activation function and all previous activation functions (for the based model the Clean and APGD accuracy are 84.59% and 49.12%, respectively).

| | Single Activation | | | | Cumulative | | | |
| | Threshold | | | | Threshold | | | |
| | 0.2 | | 0.35 | | 0.2 | | 0.35 | |
| Index | Clean | APGD | Clean | APGD | Clean | APGD | Clean | APGD |
|---|---|---|---|---|---|---|---|---|
| 0 | 84.52 | **49.34** | 84.21 | 49.19 | 84.52 | 49.37 | 84.21 | 49.19 |
| 1 | 84.56 | 49.19 | 84.44 | 49.09 | **84.53** | 49.38 | **84.31** | 49.35 |
| 2 | 84.46 | 49.21 | 84.45 | 49.20 | 84.49 | 49.37 | 83.98 | 49.29 |
| 3 | 84.50 | 49.15 | 84.33 | 49.19 | 84.45 | 49.52 | 83.72 | 49.69 |
| 4 | 84.48 | 49.21 | 84.26 | **49.26** | 84.41 | 49.67 | 83.47 | 50.02 |
| 5 | 84.54 | 49.24 | 84.36 | 49.25 | 84.38 | 49.86 | 83.14 | 50.29 |
| 6 | 84.60 | 49.11 | 84.58 | 49.21 | 84.34 | 49.90 | 83.21 | 50.50 |
| 7 | 84.51 | 49.13 | 84.45 | 49.16 | 84.32 | 49.99 | 83.02 | 50.65 |
| 8 | 84.58 | 49.19 | 84.59 | 49.21 | 84.34 | 50.12 | 83.05 | 50.86 |
| 9 | 84.60 | 49.12 | 84.53 | 49.02 | 84.35 | 50.32 | 82.96 | 51.09 |
| 10 | 84.59 | 49.06 | 84.59 | 49.07 | 84.34 | 50.35 | 83.12 | 51.15 |
| 11 | 84.60 | 49.06 | 84.56 | 49.05 | 84.38 | 50.33 | 83.02 | 51.23 |
| 12 | 84.62 | 49.09 | 84.72 | 49.05 | 84.37 | 50.27 | 83.09 | 51.23 |
| 13 | 84.59 | 49.13 | 84.63 | 49.09 | 84.38 | 50.37 | 83.14 | 51.20 |
| 14 | 84.60 | 49.17 | 84.63 | 49.12 | 84.40 | 50.34 | 83.26 | **51.41** |
| 15 | 84.57 | 49.15 | 84.69 | 49.10 | 84.36 | **50.38** | 83.30 | 51.37 |
| 16 | **84.63** | 49.09 | **84.75** | 49.01 | 84.32 | 50.37 | 83.51 | 51.36 |

Table 11: Clean and APGD accuracy of ResNet-18 model with GELU activation function over CIFAR-10 test set before (Base denoted by $\alpha = 0$ in the table) and after the application of MEANSPARSE technique for *Main Path* (application of MEANSPARSE to activation functions in the main path of residual block) and *After Addition* (application of MEANSPARSE to activation functions operating after addition of main path result and residual data) cases.

| | Layer type | | | |
| | Main Path | | After Addition | |
| $\alpha$ | Clean | APGD | Clean | APGD |
|---|---|---|---|---|
| Base | **84.59** | 49.12 | 84.59 | 49.12 |
| 0.05 | 84.59 | 49.20 | **84.62** | 49.25 |
| 0.10 | 84.59 | 49.31 | 84.59 | 49.43 |
| 0.15 | 84.50 | 49.43 | 84.49 | 49.53 |
| 0.20 | 84.40 | 49.63 | 84.47 | 49.66 |
| 0.25 | 84.26 | 49.68 | 84.46 | **49.93** |
| 0.30 | 83.94 | 49.88 | 84.27 | 49.90 |
| 0.35 | 83.66 | **49.94** | 84.15 | 49.73 |
| 0.40 | 82.96 | 49.91 | 83.67 | 49.24 |

Table 12: Clean and PGD-50 Madry et al. (2018) accuracy of various RobustBench-ranked models Croce et al. (2020), both before (Base) and after integration with the MEANSPARSE technique (models tested here includes WideResNet-94-16 Bartoldson et al. (2024) and RaWideResNet-70-16 Peng et al. (2023) over CIFAR-10 dataset, WideResNet-70-16 Wang et al. (2023) over CIFAR-100 dataset and Swin-L Liu et al. (2024) and ConvNeXt-L Liu et al. (2023) over ImageNet dataset

| | CIFAR-10 | | | | CIFAR-100 | | ImageNet | | | |
| | WideResNet-94-16 | | RaWideResNet-70-16 | | WideResNet-70-16 | | Swin-L | | ConvNeXt-L | |
| | Clean | PGD-50 | Clean | PGD-50 | Clean | PGD-50 | Clean | PGD-50 | Clean | PGD-50 |
|---|---|---|---|---|---|---|---|---|---|---|
| Base | **93.68** | 76.36 | **93.27** | 74.01 | **75.23** | 48.35 | **78.92** | 61.42 | **78.02** | 60.62 |
| MEANSPARSE | 93.63 | **78.21** | 93.24 | **75.17** | 75.17 | **50.64** | 78.88 | **61.78** | 77.96 | **61.78** |