# OpenReview forum: "MeanSparse: Post-Training Robustness Enhancement Through Mean-Centered Feature Sparsification"
_ICLR.cc/2025/Conference — Submitted to ICLR 2025_

### Official Review · Reviewer_bqUf · 2024-10-18

**Soundness:** 3
**Presentation:** 3
**Contribution:** 3
**Rating:** 3
**Confidence:** 5

**Summary:**

The paper introduces MeanSparse, a novel post-training method designed to enhance the robustness of convolutional and attention-based neural networks. MeanSparse works by sparsifying mean-centered feature vectors. This technique effectively decreases the success rate of adversarial attacks by minimizing the exploitable feature variations. The paper demonstrates that MEANSPARSE improves the robustness on datasets such as CIFAR-10, CIFAR-100, and ImageNet.

**Strengths:**

1. The application of sparsification to mean-centered features is an innovative approach, as it targets non-robust features in a unique way.

2. Authors provide comprehensive results across multiple datasets and models

3. The paper is well-organized and clearly describes the underlying motivation, methodology, and results

**Weaknesses:**

1. whether its effectiveness generalizes across different types of attention-based models, such as ViT?

2. Could you provide provide an analysis of how MEANSPARSE's effectiveness scales with model size? Authors only apply their method on large networks, like wrs-70. How about the performance of the method on smaller network, like ResNet-18, Swin-Small?

3. After reading the codes provided, I found that the MeadSquare will calculate the mean and var of the input data to get running_mean and running_var. I do no think the model should change its any parameters according to the test data.

4. The number of baselines in this paper is too few. Authors do not compare their methods with other sparsity methods.

**Questions:**

1. Could the method be used for adversarial training and accumulate the running mean and running var during training? Then during testing, fix the running mean and var.

2. Could you provide a detailed analysis of how MEANSPARSE's effectiveness changes as the attack strength increases, like 16/255?

---

> ### Author Response · Authors · 2024-11-23
> **Rebuttal by Authors**
>
> Thank you for your detailed review and valuable feedback. Below, we address your concerns:
>
> $\textbf{Generalization Across Attention-Based Models:}$
>
> Our primary objective in this paper is to compare the performance of MeanSparse-integrated models with the original adversarially trained ones. While there are several robust CNN architectures, the options for attention-based architectures are limited. Additionally, training well-robustified models is time-consuming. In the current version of the paper, we include the Swin-L transformer from attention-based architectures and ConvNeXt-L. Although ConvNeXt-L is not attention-based, it draws inspiration from the design principles of vision transformers. Our results demonstrate that MeanSparse is effective for both architectures.
>
> $\textbf{Smaller Models:}$
>
> We have conducted experiments to evaluate the effectiveness of MEANSPARSE on smaller models, such as ResNet-18, as part of our ablation studies. These experiments examined several key factors that could influence robustness. Across various settings, we demonstrated that MEANSPARSE improves robustness on smaller models as well. For a detailed analysis of the ResNet-18 results, please refer to Appendix A. Furthermore, Figure 3 presents results for models with varying parameter sizes and architectures, highlighting the effectiveness of our method across diverse model types.
>
> $\textbf{Statistics During Test:}$
>
> The running mean and variance are fixed during testing, ensuring that no parameters are changed during testing, and are computed using the training set as it provides a more reliable estimation of the mean and variance across different features.
>
> $\textbf{Other Sparsity Methods:}$
>
> MeanSparse is a post-training operator applied to feature vectors, which sets it apart from traditional sparsity methods typically used for regularization. While sparsity metrics such as the L1 norm [1] and SL0 regularization [2] could be alternatives, their proximal operators (or shrinkage functions) impact all feature values. In theory, every feature is altered based on its distance to the mean, leading to a significant decline in both clean and robust accuracy, making them unsuitable for post-training adjustments. In contrast, the hard-thresholding operator used in MeanSparse only affects features near the mean, preserving distant features that carry valuable information. This selective impact makes the L0 sparsity metric more effective in this context. Furthermore, to our knowledge, no existing research has demonstrated the use of sparsity shrinkage to enhance model robustness.
>
> [1] Tibshirani, R. (1996). Regression shrinkage and selection via the lasso. Journal of the Royal Statistical Society: Series B (Methodological), 58(1), 267–288. https://doi.org/10.1111/j.2517-6161.1996.tb02080.x
>
> [2] Mohimani, H., Babaie-Zadeh, M., Gorodnitsky, I., & Jutten, C. (2010). Sparse Recovery using Smoothed $\ell^ 0$(SL0): Convergence Analysis. arXiv preprint arXiv:1001.5073.
>
> $\textbf{Questions:}$
>
> $\textbf{1-}$ Accessing high-probability feature regions is crucial for MeanSparse. During training, as features are being learned, their statistics continuously change, making them unsuitable for our purpose. To address this, we compute the required statistics in a single additional epoch after training.
>
> The paper also discusses integrating MeanSparse during training, which requires a carefully designed threshold scheduler. If the threshold increases too quickly, it disrupts training by causing the gradient zeroing effect of MeanSparse. Conversely, if it increases too slowly, the behavior resembles post-training integration. Furthermore, aligning the threshold scheduler with the activation functions is critical. These complexities make it challenging to incorporate MeanSparse into large models during the initial training phase. We include a detailed comparison of MeanSparse integration during and post-training in Section 3.3 of the revised manuscript.
>
> $\textbf{2-}$ During our ablation study, we analyzed the effectiveness of MEANSPARSE across different attack strengths. Specifically, we measured the APGD accuracy of the ResNet-18 model with the GELU activation function on the CIFAR-10 test set before and after integrating MEANSPARSE under varying attack powers. For instance, with a threshold of 0.2, we observed that the robust accuracy was slightly lower than the base model (threshold = 0) for attack powers of ϵ=1/255 and ϵ=2/255. This decrease is primarily attributed to the reduction in clean accuracy caused by applying MEANSPARSE. However, for attack powers ϵ>2/255, MEANSPARSE consistently improved the APGD accuracy, demonstrating its effectiveness against stronger attacks. For example, at ϵ=16/255, APGD accuracy improves from 11.97% (base model) to 13.41% (threshold 0.2) and 15.44% (threshold 0.35).
>
> Please refer to Appendix A.8 and Table 9, which provide a comprehensive analysis across different attack strengths.

---

### Official Review · Reviewer_ry3U · 2024-11-04

**Soundness:** 3
**Presentation:** 3
**Contribution:** 3
**Rating:** 6
**Confidence:** 3

**Summary:**

This paper seeks to improve adversarial training by inducing sparsity in the features of an adversarially-trained model. To achieve this, MeanSparse blocks variations in features within a given distance of the mean, which is intended to lessen the importance of non-robust features during training. Before describing their sparsification method, the authors provide intuition as to how non-robust features can be removed during training through regularizad optimization. Here, a regularization term is included in the training objective which penalizes the $\ell_0$ norm of mean-centered learned features. In theory, this would remove small deviations from the mean which don't result in significant reductions in loss. Inspired by this intuition, the authors then propose a sparsification operator, which is applied during forward propagation and explicitly blocks variation within a given distance of the feature mean. Severel implementation challenges are then addressed. Experimental results show that integrating MeanSparse into state of the art image classifiers can significantly improve robustness with negligible impact on clean performance.

**Strengths:**

- The results show meaningful improvement in the robustness of SOTA adversarially trained models. Since this method can be applied post-training, it has the potential to lead to a significant jump across the board in standards for robustness to adversarial examples.
- The experimental section appears well designed. Results are presented for CIFAR-10, CIFAR-100, and Imagenet, and a representative variety of model architectures and adversarial attacks are tested.
- The method is also shown to be robust against adaptive attacks, as shown in Appendix A.4.
- I quite like the visualizations provided in Figure 1, they simply and effectively convey how MeanSparse operates.

**Weaknesses:**

- The description of the MeanSparse technique is somewhat ambiguous to me. The term "feature" is often used, but never specifically defined. What features are being used here? Are you referring to input features? Activations of the final layer? Activations of some internal layer?
- I don't entirely agree with the provided intuition for blocking minor variations around the feature mean. Based off of the provided explanation, I would expect this approach to work when minor variations are blocked around high-probability feature values. However, it's not clear to me that the feature mean would always be a high-probability point.
- Appendix A.2 does provide results looking at $\ell_2$ bounded attacks, rather than $\ell_\infty$ bounded attacks. However, I think the topic of how MeanSparse performs against different threat models does warrant additional study. Threat models have been studied in which the adversary can introduce perturbations that are unbounded in $\ell_p$ space (i.e. [1]), and it is not obvious to me whether these types of attacks would have similar impacts on the distributions of features. If the claims made in this paper are limited to $\ell_p$ bounded attacks, I think that should be made explicit.

[1] Xiao, Chaowei, Jun-Yan Zhu, Bo Li, Warren He, Mingyan Liu, and Dawn Song. "Spatially transformed adversarial examples." arXiv preprint arXiv:1801.02612 (2018).

**Questions:**

- Is the feature mean always a high-probability point? I would imagine that in certain multimodal distributions the mean is actually unlikely to occur. Might this occur in practice?
- One component of MeanSparse involves calculating the mean and standard deviations of features. Can you provide more information regarding what data is used to calculate these values? Specifically, are these statistics computed on benign or adversarially perturbed data?
- How sensitive is the feature mean to class imbalances in the training set? If classes aren't evenly balanced (assuming feature distributions are different for different classes), could that result in changes in feature means?

---

> ### Author Response · Authors · 2024-11-23
> **Rebuttal by Authors**
>
> Dear Reviewer,
>
> Thank you for your thoughtful review and for highlighting both the strengths and areas for improvement in our work. Below, we address your concerns:
>
> $\textbf{Ambiguity Regarding "Features":}$
>
> The MeanSparse operator is typically placed between the batch normalization layer and the activation function. By "feature," we refer to the output of the layer preceding the MeanSparse operator, which is usually the output of the batch normalization layer.
>
> $\textbf{Feature Mean as a High-Probability Point:}$
>
> Precisely characterizing the mode of the distribution for a specific feature in a deep architecture is a complex task due to both the input distribution and the intricate mapping within the network. However, previous studies have demonstrated that feature distributions, particularly in the deeper layers of neural networks, tend to be unimodal [1], allowing the mean value to act as a representative of the high-probability region. Consequently, variations around the mean carry less significant information about the output. Additionally, we observe that after sparsifying the features around the mean, there is almost no reduction in accuracy, further validating our assumption that the mean is a reliable representative of the high-probability region.
>
> $\textbf{Threat Models Beyond $\ell_{\infty}$ and $\ell_{2}$:}$
>
> Thank you for your insightful feedback. Our method is not restricted to $\ell_{\infty}$-bounded attacks; its effectiveness has also been demonstrated on state-of-the-art models robust to $\ell_{2}$-bounded attacks. For example, we applied MeanSparse to the WideResNet-70-16 model, which is state-of-the-art on CIFAR-10 with an $\ell_{2}$-attack budget and a radius of 0.5. By integrating MeanSparse, the clean accuracy remained nearly unchanged (95.54% to 95.49%), while robustness significantly improved (84.97% to 87.28%), highlighting its effectiveness against $\ell_{2}$-bounded attacks.
>
> You can refer to Section 4.2 and Figure 3 for results demonstrating MeanSparse's robustness under $\ell_{2}$-bounded threats. Additionally, Appendix A.2 provides a detailed analysis showing how our method enhances robustness against both $\ell_{2}$- and $\ell_{\infty}$-bounded attacks. However, for the majority of our ablation studies, we adopt an $\ell_{\infty}$-bounded threat model, as it is the most commonly reported in the research community. We will include a note in the limitations section of the revised version to explicitly state that the attacks considered in this work are bounded, addressing your concern.
>
> $\textbf{Questions:}$
>
> $\textbf{1-}$ As discussed earlier regarding weaknesses, accurately tracking the exact high-probability region is complex because it requires estimating densities using tools such as feature histograms. However, the distributions in the deeper layers of deep learning architectures are often unimodal [1], making the mean a reasonable estimate for high-probability regions. Additionally, our simulation results confirm that blocking feature variations around the mean does not significantly affect accuracy. Therefore, we utilize the feature mean in the current version of MeanSparse. More efficient versions of MeanSparse in terms of robustness could be designed in future work by more precisely analyzing the distribution or considering multidimensional distributions instead of one-dimensional ones.
>
> $\textbf{2-}$ For each model, we used the training set of the dataset on which the model was trained. We chose benign (non-adversarial) data from the training set because it provides a more reliable estimation of the mean and variance across different features compared to the validation or test sets. This ensures that the computed statistics accurately reflect the typical feature distributions.
>
> $\textbf{3-}$ MeanSparse aims to suppress features near the unconditional mean of the distribution, without incorporating class labels into its computations. In other words, treating the transformation from input to features as an implicit generator, it empirically estimates the mean and variance of each feature independently of class labels.
>
> [1] Shwartz-Ziv, R. and Tishby, N., 2017. Opening the black box of deep neural networks via information. arXiv preprint arXiv:1703.00810.

---

> > ### Comment · Reviewer_ry3U · 2024-11-27
> >
> > Thank you for your detailed response. Considering this and the responses to the other reviewers, I choose to maintain my score.

---

### Official Review · Reviewer_8mZY · 2024-11-04

**Soundness:** 2
**Presentation:** 3
**Contribution:** 2
**Rating:** 3
**Confidence:** 5

**Summary:**

The paper proposes a post-training method for enhancing the robustness of adversarially trained models. In particular, it inserts, at various layers, the MeanSparse modules which project the features (before activation) onto their mean computed over the training set if they are closer than a threshold to the mean itself. This has the goal of preventing an attacker from exploiting non-informative features to change the predicted class. In the experiments, several SOTA robust models are equipped with MeanSparse, which leaves clean performance unchanged while improving robustness.

**Strengths:**

- The proposed method is efficient, can be applied to any pre-trained model without additional training, and incurs in limited additional inference cost.

- The experiments include several architectures, datasets and threat models.

**Weaknesses:**

- The main concern is about the possible presence of gradient masking [A]. In fact, as mentioned in the paper, the MeanSparse operator induces zero gradients for the features which are projected. Since these are supposed to be the most common ones, and MeanSparse is applied in multiple points in the network, one can expect that the computed gradient might contain limited information, and thus the gradient-based attacks not work properly. While the paper tests black-box attacks, in this case the base model is (highly) robust, and the improvements given by MeanSparse are in the order of 1-3%, which might be of the same order or even smaller than the gap between white- and black-box (with standard query budget) attacks for the base model: then it is not clear that in this case this is sufficient to exclude gradient masking. A simple adaptive attack would consist in removing the projection operation when computing the gradient in the attacks, which would modify the backward pass of the model while preserving its predictions (a similar approach to BPDA [A]).

- The discussion in Sec. 3.1 seems a bit disconnected from the final approach (also, $z_{k-1}$ in Eq. (7) is not defined).

[A] https://arxiv.org/abs/1802.00420

**Questions:**

See above.

---

> ### Author Response · Authors · 2024-11-23
> **Rebuttal by Authors**
>
> Thank you for your detailed feedback and valuable suggestions. Below, we address your concerns:
>
> $\textbf{Gradient Masking Concerns:}$
>
> Thank you for your thoughtful comment. We understand the concern regarding potential gradient masking and the need to test adaptive attacks, such as BPDA, to evaluate the robustness of MeanSparse-integrated models more thoroughly. Below is our detailed response:
>
> $\bullet \textit{Potential Gradient Masking:}$
>
> The MeanSparse operator does induce zero gradients for the features it projects, which could limit the effectiveness of gradient-based attacks. To address this, we acknowledge that adaptive attacks, such as BPDA, where the projection operation is removed during gradient computation, could potentially reduce the robust accuracy. This limitation will be explicitly highlighted in the revised manuscript to ensure transparency.
>
> $\bullet \textit{Standardized Comparison with AutoAttack:}$
>
> While adaptive attacks are valuable for understanding specific weaknesses, we used AutoAttack for all evaluations to maintain a standardized and comparable metric. AutoAttack is widely used for robustness evaluation and does not assume knowledge of the defense, ensuring consistency across models before and after integrating MeanSparse. Importantly, this also aligns with how models are evaluated in RobustBench.
>
> $\bullet \textit{Preliminary Adaptive Attack Analysis:}$
>
> To explore the effect of adaptive attacks, we tested replacing the gradient of MeanSparse with an identity function (similar to BPDA). This approach, which modifies the backward pass, led to a decrease in AutoAttack accuracy. However, we recognize that further adaptive attacks (e.g., tailored BPDA) could reveal additional limitations, and we have included this discussion in Section 4.4 of the revised manuscript.
> By combining a standardized evaluation with initial adaptive attack experiments, we aim to provide a comprehensive analysis.
>
>
> $\textbf{Discussion in Section 3.1:}$
>
> In Section 3.1, we provided detailed explanations and intuition behind the proposed MeanSparse method. While Section 3.1 may initially appear disconnected from the final approach, its relevance becomes clear upon reading Section 3.2, where the connection to the exact MeanSparse formulation is established. $Z_{k-1}$ in Eq. (7) is mistakenly written instead of $\bar{a}_{k-1}$. Eq. (7) will be corrected in the revised version of the manuscript.

---

> > ### Comment · Reviewer_8mZY · 2024-12-01
> >
> > I thank the authors for the response and additional discussion.
> >
> > I think that discussing the potential limitations without a quantitative evaluation is in this case is not sufficient, given that there exist a large body of work on adaptive attacks. Moreover, in RobustBench the evaluation of AutoAttack is integrated with adaptive attacks when necessary. Thus, I'll keep the original score.

---

### Official Review · Reviewer_ik2X · 2024-11-04

**Soundness:** 3
**Presentation:** 2
**Contribution:** 2
**Rating:** 5
**Confidence:** 3

**Summary:**

This paper introduces a method called MEANSPARSE, which enhances the adversarial robustness of trained neural networks in a post-processing manner without compromising clean accuracy. The idea behind this method is to attenuate the reliance on non-robust features by modifying activation functions. Their experiments demonstrate a significant increase in the adversarial robustness of neural networks.

**Strengths:**

1. The idea of manipulating activation functions to enhance adversarial robustness is quite interesting.
2. Extensive experiments have been conducted to support the proposed method.
3. The intuition behind the idea is provided, making it easier to follow.
4. The paper includes a discussion of its limitations.

**Weaknesses:**

1. The explanation of the method in the Introduction is vague. On page 1, lines 37 and 44, you state: "In this work,..." twice, but the first sentence mentions "design of activation functions," while the second shifts to "sparsifying features to enhance robustness against adversarial attacks." These two sentences make it unclear what the key points of your work are and how investigating activation functions is related to sparsifying features. It would be better to provide a consistent expression for the main points and briefly describe the connection between them. I suggest a clear statement about your key contributions.

2. Lack of intuitive explanation of the proposed concept. On page 1, line 48, the proposed sparsity method "Mean-based Sparsification" is not well explained, and it’s followed only by a brief description of Figure 1. Without a clear explanation of the "sparsification operator", it is difficult to follow. I suggest adding a brief explanation of the "sparsification operation" you mentioned in line 90 and providing a concise mathematical definition or detailed description of how the sparsification operator works, perhaps with a simple example.

3. There are some writing errors in the paper. In Figure 2, "equation 3" appears at the end of the title, which is confusing. It would be better to remove it or rephrase the title.

**Questions:**

1. On page 6, in the first and second paragraphs, you mention two approaches and choose the second one. I am curious why you selected the second approach as your method. Is it superior? Can you provide evidence to support your choice and a brief comparison of the two approaches, highlighting the advantages and disadvantages of each?

2. In Section 4 on page 7, I did not find the experimental settings. You only mentioned that the experiments were conducted using an NVIDIA A100 GPU. For better reproducibility, it would be helpful to include detailed experimental settings. I suggest providing details such as the software versions used, hyperparameters, data preprocessing steps, and any other relevant configuration details that would allow others to replicate their experiments

---

> ### Author Response · Authors · 2024-11-23
> **Rebuttal by Authors**
>
> We appreciate your detailed review. Below, We address your concerns to enhance clarity and quality of our work:
>
> $\textbf{Vague explanation:}$
>
> Thank you for highlighting the inconsistency. We have revised the Introduction to clearly explain how MEANSPARSE integrates activation function design with feature sparsification to enhance adversarial robustness and provide a unified description of our contributions.
>
> $\textbf{Lack of intuitive explanation:}$
>
> Thank you for highlighting the need for an intuitive explanation of the "Mean-based Sparsification" method. Here, we have provided a concise explanation with a simple example and will include it in the Introduction of the revised version.
>
> $\textit{Explanation:}$
>
> The MeanSparse operator selectively suppresses variations around the mean of feature representations, effectively filtering out non-robust features. For a given feature channel, we compute the mean ($\mu$) and standard deviation ($\alpha$) over the training set. Using a tunable threshold ($Th=α⋅σ$), we block feature values that lie within $μ±Th$, replacing them with the mean value ($μ$). This operation limits minor perturbations that adversarial attacks often exploit, while preserving the informative structure of features outside this range.
>
> For instance, consider a hypothetical feature channel with a mean ($μ$) of 0.5 and standard deviation ($σ$) of 0.2. Setting $α=1$, we block values between 0.3 and 0.7, replacing them with 0.5. This simple mechanism attenuates insignificant variations, as demonstrated in Figure 1 of the paper, where we visualize how the input histogram is transformed. The blocked region corresponds to low-information variations, enhancing robustness by reducing the attacker's exploitable capacity.
>
> $\textbf{Writing errors:}$
>
> Thank you for pointing this out. We will revise the caption of Figure 2 to improve clarity.
>
> $\textbf{Questions:}$
>
> $\textbf{1-}$ The primary motivation behind MeanSparse is to block uninformative features from the model, thereby reducing the space available to potential attackers. Below is a clear comparison of the two approaches which have been included in Section 3.3 of the revised manuscript:
>
> $\bullet \textbf{During Training:}$
>
> $\textit{Advantages:}$ MeanSparse influences the model's learned representations from the start, potentially improving robustness throughout the training process.
>
> $\textit{Disadvantages:}$
> Requires a carefully designed threshold adjustment scheduler:
>
> A rapid threshold increase disrupts training due to the gradient zeroing effect.
> A slow threshold increase mimics post-training behavior.
>
> Aligning the threshold scheduler with activation functions is challenging, particularly for large models.
>
> Difficult to scale to large models due to training instabilities.
>
> $\textit{Evidence:}$ Early experiments on smaller models often resulted in unstable training and failed convergence due to misaligned thresholds.
>
> $\bullet \textbf{Post-Training:}$
>
> $\textit{Advantages:}$
>
> Statistics of the model are already established, simplifying integration.
>
> Only requires a search over alpha values, making it scalable to large models.
>
> Successfully applied to models like Swin-L, leading to a +2.56% improvement in robustness with no destabilization.
>
> $\textit{Disadvantages:}$ Cannot influence learned representations during training.
>
> $\textit{Evidence:}$ Experimental results show that post-training integration consistently improves robustness without compromising performance, even in large-scale architectures.
>
> Due to the challenges of training large models with MeanSparse, we opted for the more effective post-training integration approach.
>
> $\textbf{2-}$ Thank you for your feedback. The experimental settings used for ablation study are:
>
> Dataset: CIFAR-10
>
> Architecture: ResNet-18
>
> Optimizer: SGD (learning rate: 0.1, weight decay: 0.0005, momentum: 0.9)
>
> Number of Epochs: 200
>
> Batch Size: 256
>
> Learning rate scheduler: The initial learning rate of 0.1 is reduced by a factor of 10 at epochs 100 and 150.
>
> Best model selection: We evaluate the model at each epoch of training and select the one with the highest PGD adversarial accuracy on the test set.
>
> Adversarial training properties: 10-step PGD adversarial training [1] with respect to $\ell_{\infty}$ attacks with a radius of 8/255 and step size of 0.0078.
>
> The experiments were conducted on an NVIDIA A100 GPU, with model training taking about 6 hours and evaluation around 20 minutes per model.
>
> The experimental details are provided in Appendix A.1. We will ensure to highlight this in the main body of the paper in the revised version for better visibility. Additionally, the code is publicly available at https://anonymous.4open.science/r/MeanSparse-84B0/ to facilitate reproducibility.
>
> [1] Aleksander Madry, Aleksandar Makelov, Ludwig Schmidt, Dimitris Tsipras, and Adrian Vladu. Towards deep learning models resistant to adversarial attacks. In International Conference on Learning Representations, 2018.

---

### Author Response · Authors · 2024-11-26
**Summary of Revisions and Updates to Address Reviewer Feedback**

We sincerely thank the reviewers for their insightful feedback and for recognizing the strengths of our paper. We have carefully addressed each comment and made several revisions to enhance the clarity and quality of the manuscript. Below is a summary of the changes made in the updated version:

$\textbf{Explanation of the MeanSparse Method in the Introduction:}$

To improve clarity, we revised the Introduction to include a concise explanation of the sparsification operator, supplemented by a simple example. This addition makes the introduction more accessible and easier to follow.

$\textbf{Figure 2 Caption:}$

We revised the caption of Figure 2 to enhance clarity and better describe the visualized content.

$\textbf{Comparison of Post- and During-Training Integration of MeanSparse:}$

In Section 3.3, we outlined two approaches to integrating the MeanSparse operator. We revised this section to include a detailed explanation of the differences between the two approaches, discussing the advantages and disadvantages of each.

$\textbf{Gradient Masking Concerns:}$

In response to reviewer feedback, we updated the Limitations section (section 4.4) to address potential concerns about gradient masking and adaptive attacks. We clarified that while MeanSparse improves robustness against non-adaptive attacks, its projection operation induces zero gradients, which could reduce efficacy against white-box adaptive attacks like BPDA. We also included results from preliminary experiments using a BPDA-like approach, demonstrating a decrease in robust accuracy when replacing the MeanSparse gradient with an identity function. These updates ensure a transparent and comprehensive discussion of MeanSparse's limitations.

$\textbf{Correction in Equation 7:}$

We corrected a typo in Equation 7, where $Z_{k−1}$​ was mistakenly written instead of $\bar a_{k−1}$. This has been addressed in the revised manuscript.

---

### Meta-Review · Area_Chair_1qbp · 2024-12-23

**Metareview:**

**Summary** This work explores sparsity for adversarial robustness by proposing and evaluating a particular sparsity transform on deep representations. The MeanSparse operation is similar to (soft-)thresholding, in reducing differences about the center to zero, but distinct in maintaining values away from zero. In this case the center is set to the feature mean and a threshold hyperparameter $\alpha$ is scaled by the feature variance. This sparsity transform is applied either during adversarial training or after training and only during inference. The after training/post-training variation is favored and is implemented by calculating feature statistics over the training data without adversarial attacks. Evaluation on the standard AutoAttack benchmark shows improvement.

**Strengths**: altering activation functions is general and of interest (ik2X, ry3U) and computationally efficient (8mZY), the experiments are extensive w.r.t. architectures/datasets/threat models (ik2X, 8mZY, ry3U, bqUf), limitations are identified (ik2X), and the work is clear with an intuitive visualization and explanation help to understand the proposed operation (ik2X, ry3U, bqUf).

**Weaknesses**: the apparent robustness of the proposed operation could be explained away by gradient masking (8mZY), the explanation of the operation is vague (ik2X, ry3U), and experimental settings are not detailed (ik2X). Note that the weaknesses raised by bqUf have been discounted, although they are worthwhile points, because they were satisfactorily discussed in the response.

**Decision**: four expert reviewers choose marginal acceptance (ry3U: 6), marginal rejection (ik2X: 5), and clear rejection (8mZY: 3, bqUf: 3). Weaknesses concerning clarity or further results were addressed, and the vote for rejection by bqUf has been qualified accordingly, but the essential weakness about gradient masking and adaptive attacks (8mZY) remains. The meta-reviewer agrees with the reply by 8mZY to the rebuttal that "discussing the potential limitations without a quantitative evaluation is in this case is not sufficient, given that there exist a large body of work on adaptive attacks". The meta-reviewer therefore sides with rejection, but encourages the authors to revise their work with further study and attacks and resubmit, so that the evaluation of the proposed defense is more convincing to experts on adversarial robustness.

**Additional Comments On Reviewer Discussion:**

The authors respond to each review and provide a a summary response. All reviewers acknowledge the response, but choose to maintain their ratings. No further author-reviewer discussion took place, and no additional points were made during the final reviewer-AC discussion phase.

- Clarity: Reviewers raised issues with the introduction and its lack of example, the description of Figure 2, the distinctions between applying MeanSparse during training or after training, and an error in an equation. The revision and rebuttal fully or partially addressed these points.
- Soundness: Reviewers shared concerns about gradient masking, due to the non-differentiability of the projection operation in MeanSparse, which could provide an illusory sense of robustness. The response and revision provide more discussion of this point, but do not evaluate adaptive attacks, and do not convince the reviewers about this issue. BPDA with the identity is not an approach-specific _adaptive_ attack. Insufficient evaluation of new defense techniques is a common issue, and is a reason for the obfuscated gradients [A] paper winning an award.
- Questions: Reviewers posed specific questions, and the authors provided a response to each question. However, while the answers provided clarifications or confirmations, this information did not change the evaluation of the reviewers.

[A] Athalye et al. ICML 2018.

---

### Decision · Program_Chairs · 2025-01-22

Reject